# Optimization and Design of Balanced BPF Based on Mixed Electric and Magnetic Couplings

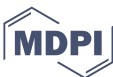

**Qiwei Li** [1,2], **Jinyong Fang** [2,*], **Wen Cao** [3], **Jing Sun** [2], **Jun Ding** [1,*], **Weihao Tie** [2], **Feng Wei** [4], **Chang Zhai** [2] **and Jiangniu Wu** [2]

1   School of Electronics and Information, Northwestern Polytechnical University, Xi'an 710129, China
2   China Academy of Space Technology (Xi'an), Xi'an 710100, China
3   School of Electronics and Control Engineering, Chang'an University, Xi'an 710064, China
4   National Key Laboratory of Antennas and Microwave Technology, Xidian University, Xi'an 710071, China
*   Correspondence: fangjy111@163.com (J.F.); dingjun@nwpu.edu.cn (J.D.)

**Abstract:** A balanced bandpass filter (BPF) with an improved frequency selectivity for differential-mode (DM) excitation and high rejection for common-mode (CM) excitation is proposed in this paper. Two half-wavelength stepped impedance resonators (SIRs) are employed based on mixed electric and magnetic couplings to realize a DM passband centered at 2.48 GHz. The center frequency and bandwidth can be easily controlled by optimizing the dimensions of SIRs and the coupling between them, respectively. Meanwhile, two transmission zeros (TZs) are generated based on the mixed electric and magnetic couplings and are independently controlled by tuning the coupling strength. Moreover, a wide DM stopband can be realized by optimizing the SIRs. The proposed balanced BPF is fed by balanced U-type microstrip–slotline transition structures, which can achieve high wideband CM rejection without influencing the DM responses, and the design complexity can be clearly reduced. Finally, a balanced BPF is fabricated, and a good agreement between the simulation and the measurement is observed, which verifies the design method.

**Keywords:** balanced; microstrip–slotline transition; mixed electric and magnetic couplings





## 1. Introduction

With the rapid development of modern wireless technologies, multiservice communication systems, such as 4G and 5G, have become widespread, and these systems will begin to interfere with each other. Recently, balanced circuits are increasingly coming into focus due to their better performance, which includes good noise immunity, low electromagnetic interference and easy connection with other balanced components or antennas compared to single-end circuits [1,2]. Balanced bandpass filters (BPFs) can achieve high common-mode (CM) rejection according to the relevant research. Several balanced BPFs with improved CM suppression were designed in [3–7]. In [3], a balanced BPF was designed by using a center-loaded half-wavelength resonator in 2010. However, its differential-mode (DM) passband selectivity and CM rejection could be improved further. In [4], a low-loss wideband differential BPF was presented by using a pair of multilayer differential microstrip–slotline transitions and microstrip line resonators. However, the proposed BPF is difficult to fabricate and expensive. A balanced filter with a broadband DM transmission and a good CM block was fabricated based on a slotline resonator in [5], but the selectivity of the DM passband could be improved further. Recently, a compact second-order balanced BPF with high selectivity was presented based on magnetically coupled resonators in [6]. However, the DM stopband is narrow. In [7], an ultra-band BPF DM passband was achieved by employing an H-type slotline structure. In order to improve selectivity, a source-load coupling was introduced. Meanwhile, one more layer of substrate was employed to reduce the DM passband insert loss. Unfortunately, the CM suppression could be improved further. In [8,9], a balanced BPF was achieved based on right-angled isosceles triangular patch

resonators and half-wavelength folded S-shaped slotline resonators, respectively. However, balanced BPFs with a compact size, good DM selectivity, low insertion loss and high CM suppression are still demanded.

A balanced BPF is designed with improved DM frequency selectivity and high CM rejection in this manuscript. The designed filter is mainly based on two half-wavelength stepped impedance resonators (SIRs), in which the SIRs are coupled by both an electric field and a magnetic field, respectively. The centered DM passband is formed at 2.48 GHz, which can be controlled by the dimensions of the SIRs. Meanwhile, two transmission zeros (TZs) are realized to achieve steep DM passband selectivity. TZs can be independently controlled by changing the coupling strength between the two SIRs. In addition, the CM suppression is broad and independent of the DM responses, which can clearly simplify the design procedure. Moreover, a broad DM stopband can also be achieved. Finally, a balanced BPF is fabricated, and a good agreement between the simulation and the measurement is observed, which verifies the design method.

## 2. Analysis and Design

### 2.1. Configuration of Balanced BPF

The traditional method to improve filter selectivity is to introduce an additional transmission path, which can generate transmission zeros. However, the size of the designed filter will increase. In this paper, the electromagnetic coupling method is used to introduce an extra transmission path while keeping the filter size unchanged. The 3D and plane structure of the designed balanced BPF is given in Figure 1. The yellow block stands for the microstrip line on the top surface of the substrate, whereas the white block and the blue block represent the slotline and ground on the bottom surface, respectively. It can be observed that the proposed component is composed of two quarter-wavelength SIRs, U-type microstrip–slotline transition structures and L-type microstrip–slotline transition structures. Generally, the design of a balanced component can be divided into two parts: DM part design and CM part design.

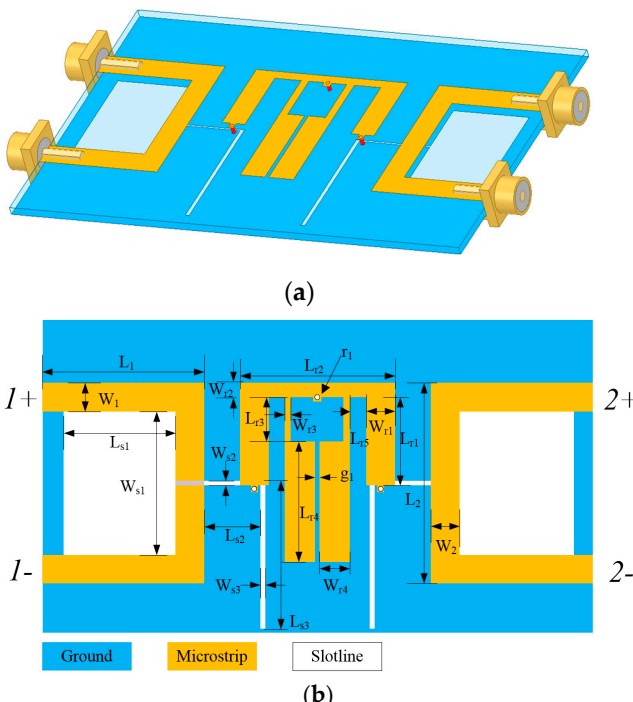

**(a)**

**(b)**

**Figure 1.** (**a**) 3D view and (**b**) top view of the proposed balanced BPF. ($W_1$ = 2.4, $W_2$ = 2.4, $L_1$ = 12.4, $L_2$ = 16, $W_{s1}$ = 11.0, $W_{s2}$ = 0.2, $W_{s3}$ = 0.4, $L_{s1}$ = 8.0, $L_{s2}$ = 5.0, $L_{s3}$ = 13.0, $L_{r1}$ = 7.0, $L_{r2}$ = 16.0, $L_{r3}$ = 5.0, $L_{r4}$ = 9.6, $L_{r5}$ = 2.8, $W_{r1}$ = 2.4, $W_{r2}$ = 1.0, $W_{r3}$ = 0.8, $W_{r4}$ = 2.6, $r_1$ = 0.2 and $g_1$ = 0.4. All values are in millimeters.)

### 2.2. DM Response

A common SIR is composed of one high-impedance section and one low-impedance section, where $Y_1$, $Y_2$ and $\theta_1$, $\theta_2$ represent the characteristic admittance and electrical length of each stub, respectively, as shown in Figure 2a. The input admittance can be concluded as [10]

$$Y_{in} = Y_2 \frac{jY_1 \tan\theta_1 + jY_2 \tan\theta_2}{Y_2 + j(jY_1 \tan\theta_1)\tan\theta_2} \tag{1}$$

$$Y_{in} = Y_2 \frac{j(Y_1/Y_2)\tan\theta_1 + j\tan\theta_2}{1 - (Y_1/Y_2)\tan\theta_1 \tan\theta_2} \tag{2}$$

For convenience, the impedance ratio is defined as $K$, and the electrical length ratio is defined as $\alpha$

$$K = \frac{Y_1}{Y_2} \tag{3}$$

$$\alpha = \frac{\theta_2}{\theta_1 + \theta_2} = \frac{\theta_2}{\theta_t} \tag{4}$$

According to the resonance condition $Y_{in} = 0$, (2) can be derived as:

$$\frac{K\tan\theta_1 + \tan\theta_2}{1 - K\tan\theta_1 \tan\theta_2} = 0 \tag{5}$$

Therefore, the resonance characteristic of SIR is determined by both impedance ratio K and length ratio $\alpha$. Beyond that, by properly tuning $K$ and $\alpha$, high order harmonics can be adjusted. Figure 2b shows a regular change in the normalized ratios of the first spurious frequency ($f_{s1}$) to the fundamental resonance frequency ($f_0$) for SIR, for which the impedance ratio $K$ increases from 0.1 to 1 when the length ratio $\alpha$ varies from 0 to 1. By adjusting impedance ratio $K$ and length ratio $\alpha$, the harmonics of SIR are far from the fundamental frequency to achieve a wide stopband feature. The initial simulation conditions are $Z_1 = 50$ Ohm, $Z_2 = 20$ Ohm, $\theta_1 = 90°$ and $\theta_2 = 60°$.

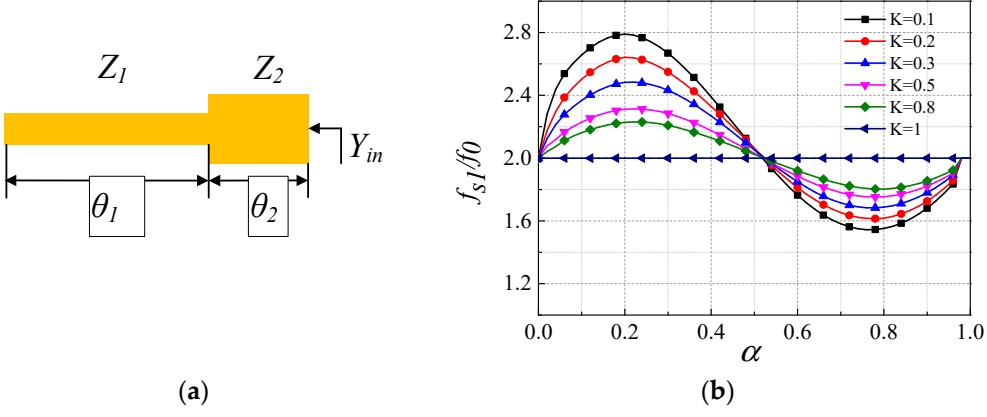

(**a**)                    (**b**)

**Figure 2.** (**a**) Configuration and (**b**) normalized ratios of the first spurious frequency to the fundamental resonant frequency of SIR.

The simplified coupling topology under DM signal excitation is shown in Figure 3, in which the blue circles, S/S′ and L/L′, represent the microstrip transmission lines with a characteristic impedance of 50 Ohm; the white circles, N and N′, denote the slotline; and the yellow circles, l and l′, represent SIRs. The coupling between the microstrip transmission line and slotline is magnetic field coupling. However, there are two kinds of couplings between the two SIRs. The first kind of coupling is electrical field coupling, which is achieved by the parallel coupling between the low-impedance sections of SIRs. The second kind of coupling is magnetic field coupling, which is realized by a metallized vie hole between the high-impedance parts of SIRs. Figure 4 shows the electrical field and magnetic

field distributions of the two SIRs at 2.47 GHz, which are achieved in EM simulation software HFSS. It can be seen that the electrical field coupling between the low-impedance parts of SIRs is strong, while the electrical field coupling between the high-impedance parts is weak at 2.47 GHz. Meanwhile, the magnetic field coupling between the high-impedance sections of SIRs is strong, while the magnetic field coupling between the low-impedance parts is weak at 2.47 GHz, which validates the aforementioned analysis. The electrical field coupling can be controlled by the gap between the low-impedance parts of SIRs, while the magnetic field coupling can be controlled by the diameter of the metallized via hole. The coupling between the slotline and microstrip line occurs by means of the magnetic field. The slotline, which is etched on one side of the substrate, is crossed at a right angle by a microstrip conductor on the opposite side. The microstrip and slotline extend about one quarter of a wavelength beyond the crossing point. To reduce the size, the extended quarter-wavelength open-circuit branch can also be replaced by a short-circuit point.

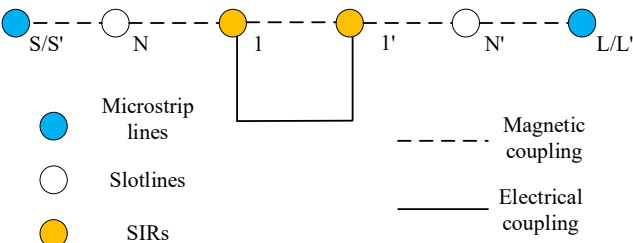

**Figure 3.** Simplified coupling topology under DM signal excitation of the proposed filter.

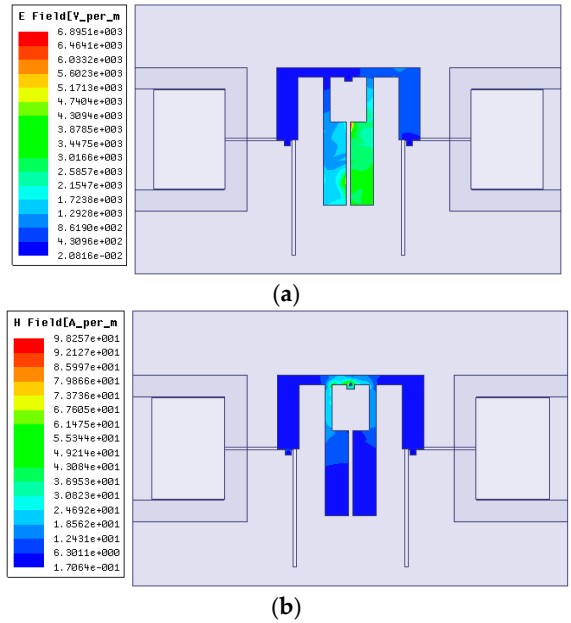

**Figure 4.** (**a**) Electrical field and (**b**) magnetic field distributions of the two SIRs at 2.47 GHz.

The performances of the DM and CM frequency characteristics of the proposed balanced filter with different dimensions are also analyzed by HFSS 13.0. As observed in Figure 5, the center frequency, bandwidth and the positions of $TZ$ can be controlled by adjusting the dimensions. As observed in Figure 5a,b, the center frequency of the proposed balanced BPF is decreased when $L_{r4}$ is increased, and the bandwidth is decreased when $g_1$ is increased. For the coupling resonators, the electrical coupling and magnetic coupling can be achieved by the gap between the high-impedance sections of SIRs and the metallized via hole. Due to the cancellation effect of the electrical coupling and magnetic coupling, $TZ_l$ (the first transmission zero) is realized in the lower passband. Due to the U-type microstrip–slotline transition structure, $TZ_h$ (the second transmission zero) is generated

in the upper passband. Therefore, the selectivity of the DM passband can be improved greatly. As shown in Figure 5c, the position of $TZ_l$ is controlled by changing $r_1$. Meanwhile, $TZ_h$ is moved to high frequency when $L_{r5}$ increases, as observed in Figure 5d. In addition, step impedance slotline structure is used to achieve the miniaturization of the proposed balanced BPF. Additionally, the impact of slotlines' width and length on the transmission performance is shown in Figure 5e. Therefore, the DM performances of the proposed balanced filter can be independently controlled by the related parameters.

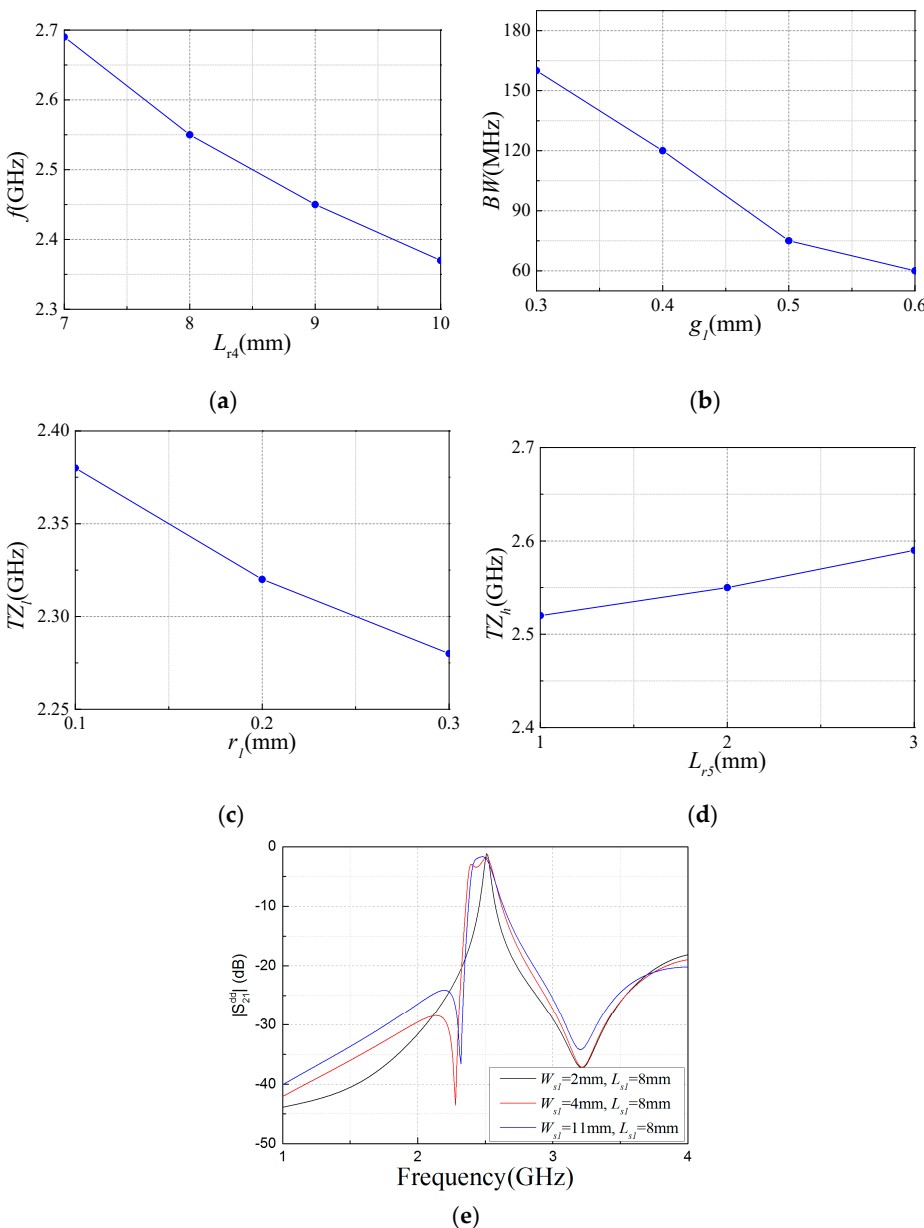

**Figure 5.** DM responses with different parameters: (**a**) $L_{r4}$; (**b**) $g_1$; (**c**) $r_1$; (**d**) $L_{r5}$; (**e**) $W_{s1}$. ($W_1$ = 2.4, $W_2$ = 2.4, $L_1$ = 12.4, $L_2$ = 16, $W_{s1}$ = 11.0, $W_{s2}$ = 0.2, $W_{s3}$ = 0.4, $L_{s1}$ = 8.0, $L_{s2}$ = 5.0, $L_{s3}$ = 13.0, $L_{r1}$ = 7.0, $L_{r2}$ = 16.0, $L_{r3}$ = 5.0, $L_{r4}$ = 9.6, $L_{r5}$ = 2.8, $W_{r1}$ = 2.4, $W_{r2}$ = 1.0, $W_{r3}$ = 0.8, $W_{r4}$ = 2.6, $r_1$ = 0.2 and $g_1$ = 0.4. All values are in millimeters.)

### 2.3. CM Suppression

A slotline is a planar transmission structure first proposed by Cohn in 1968, which consists of a narrow slot etched in the metallization on one side of a dielectric substrate. The metal on the other side of the substrate is removed. This geometry is well suited for applications in microwave-integrated circuits. The slotline, which is etched on one side

of the substrate, is crossed at a right angle by a microstrip conductor on the opposite side. The microstrip extends about one quarter of a wavelength beyond the slot. The transition can be fabricated using the usual photoetching process and is thus easily reproducible. Additionally, as the microstrip part of the circuit can be placed on one side of the substrate and the slotline part on the other side, this transition makes two-level circuit design possible. Coupling between the slotline and microstrip line occurs by means of the magnetic field. A balanced U-type microstrip/slotline transition, as depicted in Figure 6, comprises a U-type stepped-impedance microstrip feedline and a slotline resonator. The slotline resonator is etched on the bottom ground and the U-type microstrip line is on the top side of the substrate. The width and length of the slotline will influence the conversion efficiency from microstrip line to slotline, i.e., the insertion loss of the bandpass filter. Generally, the slotline extends about one quarter of a wavelength beyond the intersection of microstrip lines and slotlines. The electric field distributions excited by DM and CM signals in U-type microstrip–slotline transition structure are shown in Figure 6, in which the red arrow represents the electric field in microstrip line, the purple arrow represents the electric field in slotline and the symmetry plane is denoted by A-A'. When the balanced ports are excited by DM signals, the symmetric plane, A-A', can be regarded as an electrical wall, as observed in Figure 6a. The DM signals along the U-type microstrip line can be converted easily into the slotline mode because of the strong magnetic coupling. However, when the circuit is under CM signal excitation, a virtual magnetic wall will be realized at the plane, A-A', as observed in Figure 6b. Because the electric field of the slotline and the magnetic wall are vertical to each other, the CM signals cannot be coupled to the slotline. Moreover, the CM characteristic of the U-type microstrip–slotline transition structure is related to its own boundary condition and independent of the working frequency, as observed in Figure 7. Therefore, the design complexity can be reduced significantly. To reduce the size, the slotlines are designed as stepped impedance, and the specific dimensions are optimized by HFSS. The uniform-impedance slotline and the stepped-impedance slotline can both create good CM suppression. The length of the balanced microstrip–slotline transition structure can be effectively reduced by employing stepped-impedance slotline. Therefore, a stepped-impedance slotline structure is used to achieve the miniaturization of the proposed balanced BPF.

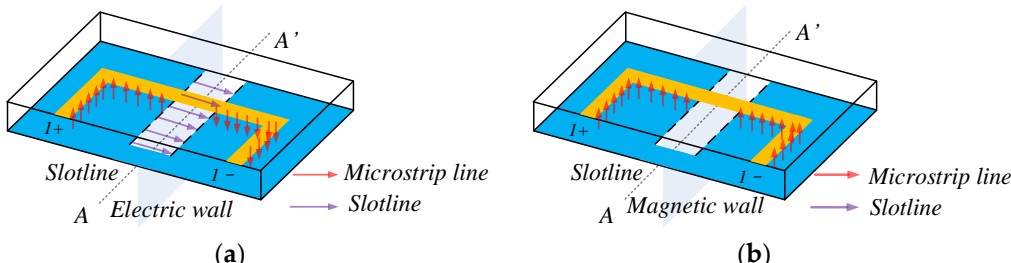

**Figure 6.** Electric field distributions of the balanced feed ports under (**a**) DM and (**b**) CM excitation.

### 2.4. Design Procedure

Based on the above analysis and discussions, a simple design procedure of the proposed balanced BPF can be summarized as follows:

Step (1): For the desired center frequency, the lengths of SIR can be selected appropriately according to Figure 5a.

Step (2): For the desired bandwidth, other dimensions of SIR can be selected appropriately according to Figure 5b–d.

Step (3): For the desired center frequency, the lengths of the slotlines can be chosen.

Step (4): All values are further optimized to realize better DM and CM responses of the proposed BPF. Full-wave electromagnetic simulation and dimension optimization in the commercial software of HFSS can be conducted.

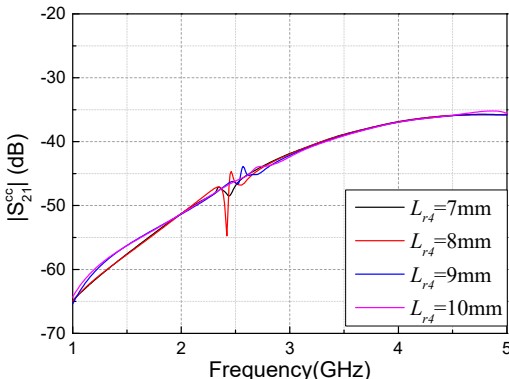

**Figure 7.** Simulated frequency responses with CM excitation of the proposed balanced filter for different $L_{r4}$. ($W_1$ = 2.4, $W_2$ = 2.4, $L_1$ = 12.4, $L_2$ = 16, $W_{s1}$ = 11.0, $W_{s2}$ = 0.2, $W_{s3}$ = 0.4, $L_{s1}$ = 8.0, $L_{s2}$ = 5.0, $L_{s3}$ = 13.0, $L_{r1}$ = 7.0, $L_{r2}$ = 16.0, $L_{r3}$ = 5.0, $L_{r5}$ = 2.8, $W_{r1}$ = 2.4, $W_{r2}$ = 1.0, $W_{r3}$ = 0.8, $W_{r4}$ = 2.6, $r_1$ = 0.2 and $g_1$ = 0.4. All values are in millimeters.)

## 3. Simulation and Measurement

The photographs of the fabricated balanced BPF are shown in Figure 8. The substrate used in this design is F4BM ($\varepsilon_r$ =2.2 and $h$ = 0.8 mm). Due to the machining accuracy, there is a certain deviation between the simulation and the measurement. The effective size is about 43.4 mm × 20.9 mm ($0.49\lambda_g$ × $0.24\lambda_g$, where $\lambda_g$ is the waveguide wavelength at 2.48 GHz). Figure 9 shows the simulation and the measurement. The measured S-parameters are carried out with an Agilent Network Analyzer N5230A. The measured DM center frequency is 2.48 GHz, with a 3-dB fractional bandwidth (FBW) of 4%. The measured minimum DM insertion loss for the center frequency is 1.74 dB, and the return loss is greater than 14 dB. Moreover, the DM stopband is extended to 10.8 GHz with a rejection level of 11.5 dB, which is 4.35 times the fundamental frequency, as shown in Figure 10. For the CM responses, a suppression of more than 40 dB can be achieved from DC to 6 GHz. Table 1 shows the comparison of the proposed balanced BPF with other proposed BPFs. It can be observed that the fabricated balanced BPF shows a better frequency selectivity of the DM passband and a higher CM rejection.

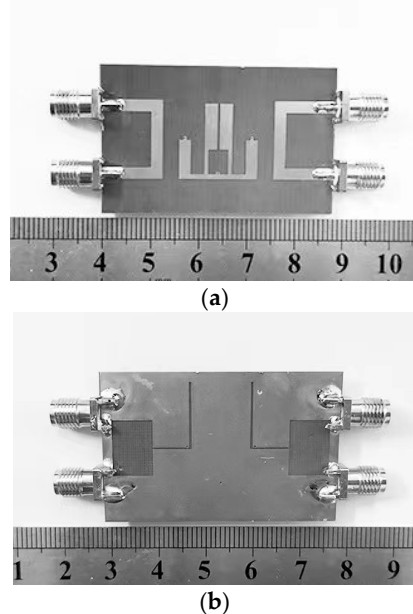

**Figure 8.** Photographs of the fabricated balanced BPF: (**a**) Top view; (**b**) bottom view.

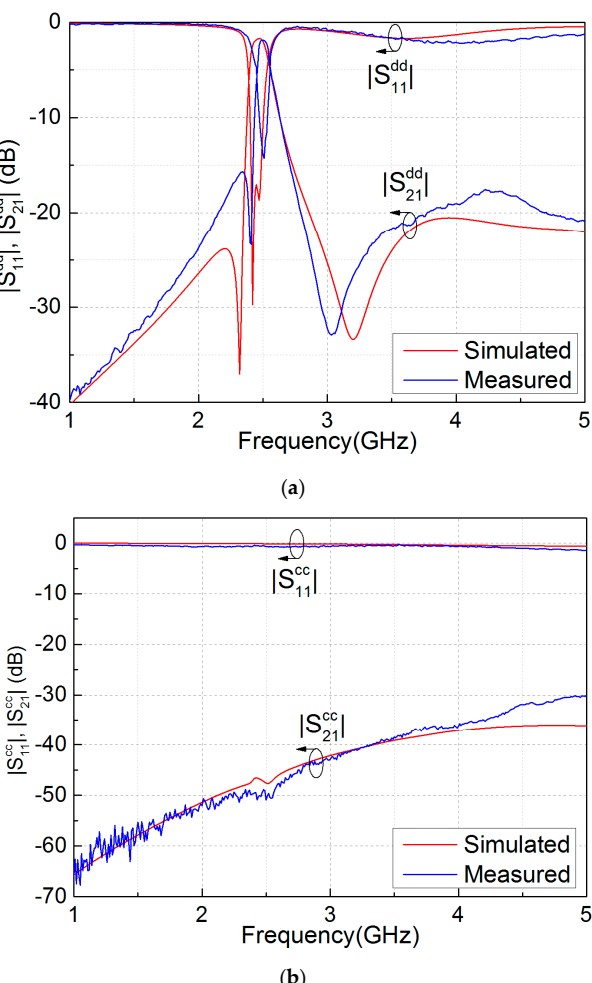

(a)

(b)

**Figure 9.** Comparisons between the simulated and measured results (narrowband): (**a**) DM responses; (**b**) CM responses. ($W_1 = 2.4$, $W_2 = 2.4$, $L_1 = 12.4$, $L_2 = 16$, $W_{s1} = 11.0$, $W_{s2} = 0.2$, $W_{s3} = 0.4$, $L_{s1} = 8.0$, $L_{s2} = 5.0$, $L_{s3} = 13.0$, $L_{r1} = 7.0$, $L_{r2} = 16.0$, $L_{r3} = 5.0$, $L_{r4} = 9.6$, $L_{r5} = 2.8$, $W_{r1} = 2.4$, $W_{r2} = 1.0$, $W_{r3} = 0.8$, $W_{r4} = 2.6$, $r_1 = 0.2$ and $g_1 = 0.4$. all values are in millimeters.)

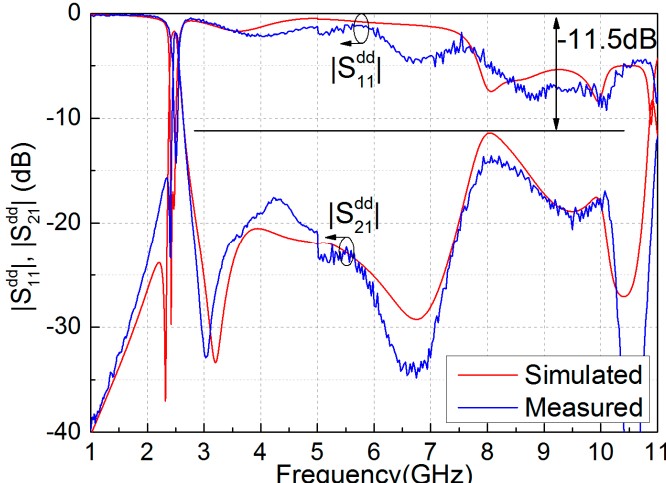

**Figure 10.** Comparisons between the simulated and measured results (wideband). ($W_1 = 2.4$, $W_2 = 2.4$, $L_1 = 12.4$, $L_2 = 16$, $W_{s1} = 11.0$, $W_{s2} = 0.2$, $W_{s3} = 0.4$, $L_{s1} = 8.0$, $L_{s2} = 5.0$, $L_{s3} = 13.0$, $L_{r1} = 7.0$, $L_{r2} = 16.0$, $L_{r3} = 5.0$, $L_{r4} = 9.6$, $L_{r5} = 2.8$, $W_{r1} = 2.4$, $W_{r2} = 1.0$, $W_{r3} = 0.8$, $W_{r4} = 2.6$, $r_1 = 0.2$ and $g_1 = 0.4$. All values are in millimeters.)

**Table 1.** Performance comparison with some published balanced BPFs.

| Refs. | $f_0$(GHz)/3-dB FBW (%) | Number of TZs | CMS (dB) ※ | Upper Stopband (GHz) | Location of TZs (GHz) |
|---|---|---|---|---|---|
| [5] | 2.4/4 | 2 | 35 | 5.0 | 0/8.5 |
| [8] | 2/12.9 | 2 | 27.8 | 4.0 | 1.01/2.62 |
| Ref. [9] Design I | 1.2/20.1 | 1 | 55 | 2.0 | 0/1.75 |
| Ref. [11] Case 2 | 6.85 | 82 | 0 | 40 | None |
| This work | 2.47/6.4 | 2 | 45 | 10.8 | 2.4/3.01 |

※ CMS = Common-mode suppression.

## 4. Conclusions

A balanced BPF based on a U-type microstrip–slotline transition structure and mixed electric and magnetic couplings is designed in this paper. The measured minimum DM insertion loss for the center frequency is 1.74 dB, and the return loss is greater than 14 dB. Two TZs located at 2.4 and 3.01 GHz are realized by introducing the electric coupling and the magnetic coupling into the design to achieve the steep DM passband selectivity. Moreover, the DM stopband is extended to 10.8 GHz with a rejection level of 11.5 dB, which is 4.35 times the fundamental frequency. The CM suppression level can be up to 40 dB from DC to 6 GHz. The designed balanced BPF exhibits good DM passband performance and wideband CM rejection, which will be valuable in fully balanced RF front ends.

**Author Contributions:** Q.L., J.F., W.C., J.S., W.T., J.D., C.Z. and J.W. designed and fabricated the proposed filter. Q.L. and F.W. contributed to the discussion and reviewed the manuscript. F.W. revised the manuscript. All authors have read and agreed to the published version of the manuscript.

**Funding:** This work was supported by the Foundation Enhancement Plan, the National Key Laboratory of Science and Technology on Space Microwave (No: 6142411332211), the National Natural Science Foundation of China (61803042) and the Fundamental Research Funds for the Central Universities, CHD (300102322103).

**Data Availability Statement:** The data presented in this study are available on request from the corresponding author.

**Conflicts of Interest:** The authors declare no conflict of interest.

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
