# Peer review of "Optimization and Design of Balanced BPF Based on Mixed Electric and Magnetic Couplings"

_electronics, doi:10.3390/electronics12092125_

Round 1

Reviewer 1 Report

Manuscript ID: electronics-2281119

Optimization and design of Balanced BPF Based on Mixed Electric and Magnetic Couplings

Authors presented the detailed analysis, design and optimization of BPF using mixed mode coupling. The article is important and needed for many RF/microwave application. The fabrication and characterization of balanced BPF proves the precision of design. The paper is good and worthy. It may need revisions. Few comments are listed below, up on review of the manuscript, which may be considered for revision by authors.

Major Comments:

1.      The introduction presents the review of balanced BPF designs, reported elsewhere. But the demands and requirements of RF/microwave application, say 5G systems, are missing.

2.      Also, the introduction lists the designs from early reports. It may be good if the performance metrics of BPFs are included, such as selectivity, isolation, filter size, passband ripples, group delay etc.

3.      In section-2, the analysis and design, should start either with an application note or the novelty of design strategy. Alternatively, authors introduce the filter in the first place, then the coupling elements are studied. Authors shall decide on this matter.

4.      Other than coupling elements, there are many structures, such as slots and shorting pins, notices. But they are not elaborated.

5.      In the caption of Fig-1 lists all dimensions of the structure. Can you give the calculation of those dimensions?

6.      Table-1 needed few more rows to compare the performance

7.      What is the use of theory presented in page-3? Does this add value to the present work? Is there any justification in the later part of the article?

8.      In overall, the optimization of the filter is not performed. Also, the design has no critical objective. Author may keep the readers in mind while writing the article.

Reviewer 2 Report

- For the reader, the presentation of the formula sequence (Eq.1-4) is insufficient. For example, the definition of Y1, Y2 but also the derivation of Eq.(4) are missing.

- To follow the simulation results presented in Figures 2b, 5(a-e), 7, 9(a,b), and 10 appropriate equation formalism is needed, with all the conditions used in the simulations.

- Looks strange when the references [9] and [10] provided in the Table only, as for comparison, are not presented as state-of-the-art in the Introduction.

- The deviation of the simulated results from the measured are not discussed, why it was impossible by adjusting the parameters to get a better fit of the experimental curves?

- The TZs frequencies should be provided in Table 1.

- The Conclusions are only qualitative, the information on competitive achievements in the work is missing.

Technical:

- The legends in Figure 4 are too small for reading.

- Figures 8(a and b) should be enlarged and of better quality.

Reviewer 3 Report

This gives the results on a microstrip band pass filter centered at about 2.3GigaHertz using both electrical and magnetic coupling. It gives the structural layout of the filter with experimental results, the latter being compared with the simulations. As such it would appear to have publishable material. However: 1. Equation (1) uses Yi whereas K uses Zi. It would probably be better to give K in terms of Yi, 2, (1) is said to come from reference 8 but looking at reference 8 the equation differs. So an explanation of how the equation in ref. 8 is turned into (1) or a derivation of (1) should be given. 3. One is actually interested in the S21 band-pass characteristics as shown in Fig. 5. So a derivation of this should be given and in particular the center frequency and bandwidth in terms of the microstrip/slotline layout parameters as needed for a design. 4. An explanation as to the design of the slotline portion would seem to be required. 5. There also should be a data statement and how a reader obtains the simulation program listing so that a reader may be able to reproduce the results in the paper. 

Round 2

Reviewer 1 Report

Authors responded all comments and edited the manuscript. It looks the article is qualified for publication. Few spell check is required, such as Ohm (O should be capital). 

Reviewer 2 Report

Comment 2:

To follow the simulation results presented in Figures 2b, 5(a-e), 7, 9(a,b), and 10 appropriate equation formalism is needed, with all the conditions used in the simulations.

Response to comment 2:

Thank for your suggestion. The title of Figure 1 shows the detailed dimension of the proposed filter: Figure 1. (a) 3D view and (b) top view of the proposed balanced BPF. (W1=2.4, W2=2.4, L1=12.4, L2=16, Ws1=11.0, Ws2=0.2, Ws3=0.4, Ls1=8.0, Ls2=5.0, Ls3=13.0, Lr1=7.0, Lr2=16.0, Lr3=5.0, Lr4=9.6, Lr5=2.8, Wr1=2.4, Wr2=1.0, Wr3=0.8, Wr4=2.6, r1=0.2, all are in millimeters)

Comment to the Response:

Not answered.

Comment 4:

The deviation of the simulated results from the measured are not discussed, why it was impossible by adjusting the parameters to get a better fit of the experimental curves?

Response to comment 4:

Thank for your suggestion. In the simulation in HFSS, the substrate used is Rogers 5880 (εr =2.2 and h=0.8 mm). However, the actual substrate processed is F4BM (εr =2.2 and h=0.8 mm) made in China. Due to the different plates and machining accuracy, there is a certain deviation between simulation and measurement. The descriptions of the substrate used in this design have been updated in Introduction of the resubmitted manuscript:

“The photographs of the fabricated balanced BPF are shown in Figure 8. The sub-strate used in this design is F4BM (εr =2.2 and h =0.8 mm). Due to the machining accuracy, there is a certain deviation between simulation and measurement.”

Comment to the Response:

Why not readjust the simulations after comparing them with the measurement result and by this extract the weak points of the machining inaccuracy? That’s the way towards improvement.

Comment 7:

The legends in Figure 4 are too small for reading.

Response to comment 7:

Thank for your suggestion. We have improved the legend in Figure 4 in the resubmitted manuscript.

Comment to the Response:

Not answered. The font of the legend is left unchanged.

Comment 8:

Figures 8(a and b) should be enlarged and of better quality.

Response to comment 8:

Thank for your suggestion. Figures 8(a and b) have been enlarged in the resubmitted manuscript.

Comment to the Response:

The brightness of the images should be increased.
